# An artificial intelligence-based approach to identify volume status in patients with severe dengue using wearable PPG data

Ngan Nguyen Lyle[1☯*], Ho Quang Chanh[1☯*], Hao Nguyen Van[2,3], James Anibal[4], Stefan Karolcik[5], Damien Ming[6], Giang Nguyen Thi[1], Huyen Vu Ngo Thanh[1], Huy Nguyen Quang[1], Hai Ho Bich[1], Khoa Le Dinh Van[1], Van Hoang Minh Tu[1], Khanh Phan Nguyen Quoc[1], Huynh Trung Trieu[3], Qui Tu Phan[3], Tho Phan Vinh[3], Tai Luong Thi Hue[3], Pantelis Georgiou[5], Louise Thwaites[1,7], Sophie Yacoub[1,7] on behalf of the Vietnam ICU Translational Applications Laboratory (VITAL) investigators¶

1 Oxford University Clinical Research Unit, Ho Chi Minh City, Vietnam, 2 University of Medicine and Pharmacy, Ho Chi Minh City, Vietnam, 3 Hospital for Tropical Diseases, Ho Chi Minh City, Vietnam, 4 Department of Engineering Science, University of Oxford, Oxford, United Kingdom, 5 Centre for Bio-Inspired Technology, Imperial College London, London, United Kingdom, 6 Centre for Antimicrobial Optimisation, Imperial College London, London, United Kingdom, 7 Centre for Tropical Medicine and Global Health, University of Oxford, Oxford, United Kingdom

☯ These authors contributed equally to this work.
¶ Membership of the Vietnam ICU Translational Applications Laboratory (VITAL) investigators is listed in the Acknowledgments.
* ngan.lyle@ubc.ca (NNL), chanhhq@oucru.org (HQC)

## Abstract

Dengue shock syndrome (DSS) is a serious complication of dengue infection which occurs when critical plasma leakage results in haemodynamic shock. Treatment is challenging as fluid therapy must balance the risk of hypoperfusion with volume overload. In this study, we investigate the potential utility of wearable photoplethysmography (PPG) to determine volume status in DSS. In this prospective observational study, we enrolled 250 adults and children with a clinical diagnosis of dengue admitted to the Hospital for Tropical Diseases, Ho Chi Minh City. PPG monitoring using a wearable device was applied for a 24-hour period. Clinical events were then matched to the PPG data by date and time. We predefined two clinical states for comparison: (1) the 2-hour period before a shock event was an "empty" volume state and (2) the 2-hour period between 1 and 3 hours after a fluid initiation event was a "full" volume state. PPG data were sampled from these states for analysis. Variability and waveform morphology features were extracted and analyzed using principal components analysis and random forest. Waveform images were used to develop a computer vision model. Of the 250 patients enrolled, 90 patients experienced the predefined outcomes, and had sufficient data for the analysis. Principal components analysis identified four principal components (PCs), from the 23 pulse wave features. Logistic regression using these PCs showed that the empty state is associated with PCs 1 ($p = 0.016$) and 4 ($p = 0.036$) with both PCs denoting increased sympathetic activity. Random forest showed that heart

**Data availability statement:** Data cannot be shared publicly because of patient confidentiality. Data are available upon reasonable request, as long as this meets local ethics and research governance criteria, from the Data Access Committee at OUCRU. The email address is dac@oucru.org.

**Funding:** This research was supported by The Wellcome Trust 215010/Z/18/Z awarded to P.G., L.T. and S.Y. The funders had no role in study design, data collection and analysis, decision to publish, or preparation of the manuscript.

**Competing interests:** The authors have declared that no competing interests exist.

rate and the LF-HF ratio are the most important features. A computer vision model had a sensitivity of 0.81 and a specificity of 0.70 for the empty state. These results provide proof of concept that an artificial intelligence-based approach using continuous PPG monitoring can provide information on volume states in DSS.

## Author summary

Dengue is a globally important public health threat with an estimated 100 million symptomatic infections occurring each year. Dengue shock syndrome (DSS) is a complication of disease that occurs when progressive hypovolemia leads to circulatory collapse. Treatment of DSS is supportive, requiring careful fluid management over a 24-to-48-hour critical phase depending on patients' volume status. However, a reliable way to assess volume status is lacking. In this study, we show that photoplethysmography (PPG) monitoring for 24 hours reflects volume status in patients with DSS. We predefined two opposite volume states - "empty" or "full" based on clinical events then compare their corresponding PPG signals. Using machine learning we showed that the empty state signal is characterized by a phenotype of sympathetic dominance. This is a novel finding that highlights the underlying pathophysiology of DSS. Furthermore, a computer vision model, using the PPG signal images as inputs, had a sensitivity of 81% in detecting the empty state. We expect that with refinement, artificial intelligence-based models applied to continuous PPG monitoring will enhance the way that acute DSS is treated, enabling dynamic and personalized fluid management.

## Introduction

Dengue is a globally important public health threat with an estimated 100 million symptomatic infections occurring each year [1]. The disease exerts a significant burden in low-and-middle-countries (LMIC) where it is endemic. Dengue has recently spread to new areas including regions in Southern Europe and the Southern United States, driven by increased global mobility and climate change [2–4]. Dengue infection has a wide clinical spectrum, ranging from asymptomatic infection to life-threatening disease complicated by severe vascular leakage leading to shock, bleeding and/or organ impairment. Currently, there are no effective antivirals or definitive therapeutics to treat dengue. Close monitoring, supportive care and fluid management are the cornerstones of treatment [5].

Dengue shock syndrome (DSS) caused by vascular leak usually occurs around the time of defervescence. During this time judicious fluid management is required for 24–48 hours. The goal is to maintain adequate perfusion in the setting of intra-vascular fluid loss, balancing the need for haemodynamic stability against the risk of iatrogenic volume overload. Current guidelines recommend that serial haemato-crits and frequent clinical assessments be used to assess a patient's response to

treatment and to guide ongoing fluid management [6]. While changes in serial haematocrits indirectly reflect leakage severity, the precise relationship between haematocrit values and vascular leakage remains unclear. Moreover, haematocrit values are less reliable in patients who are receiving parenteral fluid. Clinical signs and symptoms often occur late, close to the onset of circulatory collapse. As a result, patients with severe ongoing vascular leakage who are at risk of further episodes of shock often require invasive monitoring with arterial or venous lines. However, these procedures can result in bleeding complications, due to the profound thrombocytopenia that is typically seen with severe dengue [6]. Thus, there is an urgent need for a continuous, non-invasive, accessible and more reliable tool to assist in the monitoring of DSS patients.

Photoplethysmography (PPG) is a well-established, inexpensive and portable technology [7–9] that has been used to determine blood pressure, respiratory rate and haematocrit [10]. In our group, we have shown that a novel PPG-derived parameter, the compensatory reserve index (CRI), can predict the development of recurrent shock in DSS patients before any signs of hypotension [11]. However, the CRI is a proprietary measurement that was developed using a heamorrhagic shock model. By using low-cost wearables and directly interrogating the PPG signal, we aim to develop more transparent dengue-specific algorithms suitable for deployment in LMIC.

We therefore conducted a prospective study to explore the relationship of PPG signals with haemodynamic states in DSS patients. Our hypothesis is that continuous PPG signals can be used to classify the haemodynamic volume status of patients during DSS.

## Methods

### Study population and procedures

This study was a prospective observational cohort study performed at the Hospital for Tropical Diseases (HTD), Ho Chi Minh City, Vietnam during 2020–2022. Ethics approvals were obtained from the Oxford Tropical Research Ethics Committee (OxTREC) and at the HTD. Written informed consent was obtained from participants and/or parents/guardians.

Individuals admitted to the HTD aged ≥ 5 years old with a clinical diagnosis of dengue were screened for enrolment within 48-hours of admission in the adult and pediatric intensive care units, the emergency department and the inpatient wards. Upon enrolment, continuous PPG recording was initiated and continued over a 24-hour period. Over the same period, clinical information including patient examination, laboratory tests, fluid management and shock events were also captured using standardized case report forms. Patients were followed daily until discharge or up to 5 days from enrolment.

PPG monitoring was performed using SmartCare wrist-worn pulse oximeters (SmartCare analytics, United Kingdom). The data were sampled at a frequency of 100Hz and wirelessly transmitted via Bluetooth to bedside Android tablets. Some patients had more than one recording over the 24-hour collection period when recordings were interrupted and had to be restarted.

### Clinical outcomes

The clinical outcomes were predefined as "empty" or "full" states representing opposing haemodynamic volume states. These definitions were developed for this study based on previous work done by our group [11], showing that the 2-hour period leading up to a diagnosis of shock during the critical phase likely represents the lowest volume state, while the 2-hour period about one hour after an eligible fluid initiation event (as defined below) is likely to be closest to the highest volume state.

- An "empty" state was predefined as the two-hour period before a shock event. A shock event was an episode of clinical shock (a narrow pulse pressure of <=20 mmHg and/or hypotension for age and poor peripheral perfusion) occurring at least 6 hours after a prior shock event.

- A "full" state was predefined as the two-hour period one to three hours following an eligible fluid initiation event. Eligible fluid initiation events are fluid initiation events (either colloid or crystalloid fluid) that are not followed by any other fluid initiation event or recurrent shock event within 6 hours. Eligible fluid events are meant to capture fluid events that lead to clinical improvement and were defined in this way to avoid capturing refractory shock states in which patients have ongoing clinical shock despite fluid infusion.

Note that while fluid initiation events are sometimes precipitated by either initial or recurrent shock events, fluid initiation events do not always follow a shock event. The two event types are not paired, and they do not overlap in time. Examples of patient clinical courses showing how the clinical states were defined are shown in S1 Fig.

### Event-PPG matching and data processing

Clinical events, either shock or an eligible fluid initiation event, were aligned with the PPG waveform data in time. Event-PPG matches were identified when there was at least 5 minutes of overlap between clinical states, defined above, and the PPG signal. A single patient may have multiple event-PPG matches including more than one full or empty state or both types of states. Altogether we identified 162 event-PPG matches in 115 unique patients.

A schematic of the data processing and analysis steps is shown in Fig 1. The matched signals were filtered through a 0.5 to 10 Hz bandpass Chebyshev filter. Then, three hundred PPG segments of 20 seconds duration were sampled with replacement (i.e., bootstrap sampling) from the signal for each event-PPG match (resulting in 48,600 segments). After this, segments were assessed for quality in a stringent multi-step procedure. Segments were discarded based on device error codes, measures of skewness and kurtosis and the presence of incomplete beat cycles. Ultimately, 6639 segments (~13%) from 90 unique patients were retained for feature extraction.

### Feature extraction

Pulse wave features include both heart rate variability and waveform morphology characteristics. Features were calculated using the fiducial points of the waveform in python. We used four fiducial points - the onset, the systolic rise, the

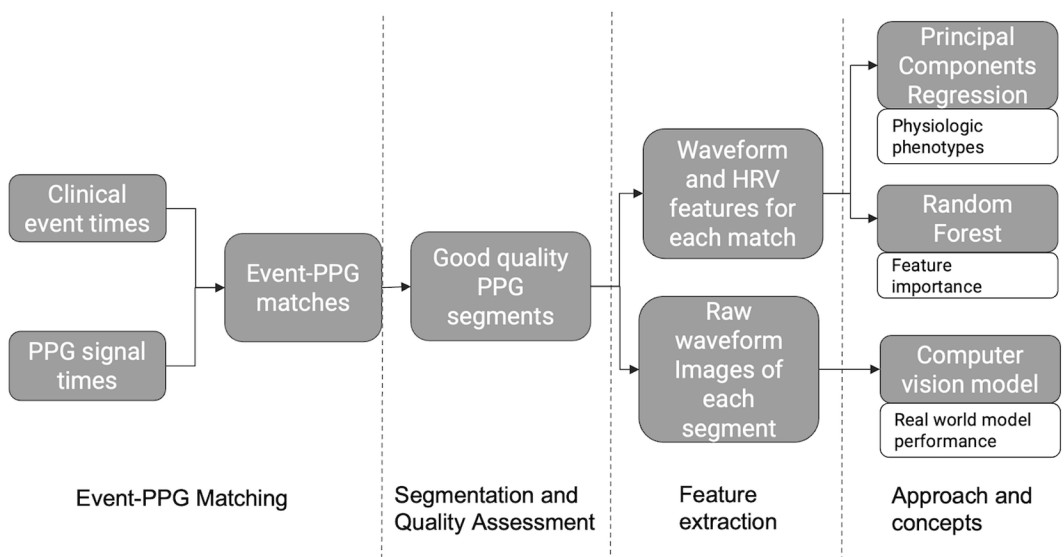

**Fig 1. Schematic of PPG data processing and analysis steps.** Abbreviations PPG – photoplethysmography, HRV – heart rate variability.

systolic peak and the dicrotic notch points which were identified using functions adapted from Charlton et. al. in python (https://peterhcharlton.github.io/bsp-book/tutorial/notebooks/pulse-wave-analysis.html). Note that this sequence of four points was defined as a complete beat cycle and only segments containing consecutive complete beat cycles were kept for analysis. Altogether, there were 23 features extracted. Features were selected based on availability and interpretability. Features involving or that could be impacted by the absolute amplitudes of the beats (e.g., height of the systolic peak or area under the beat) were excluded because absolute amplitude values are susceptible to artefact and vary depending on skin tone, probe fit and probe positioning. However, the variation in these features (e.g., the standard deviation of the heights of the systolic peaks) were included. A description of the features is shown in S1 Table. A diagram showing how the fiducial points and waveform features appear on the waveform is shown in S2 Fig. As shown in S2 Table, many of the features are highly correlated.

Since multiple beats make up a PPG segment, waveform features were determined for each beat within the segment and the mean value for each feature was taken to be representative of each segment. The standard deviation of the feature values showed the variation of these features for each segment. Heart rate variability features for each segment were determined based on the N-N intervals (see S2 Fig) for each segment, using the HRV analysis package in python [12].

## Statistical analysis

The data were reduced so that each observation is independent, belonging to a unique patient. Since a single patient can have multiple event-PPG matches, only one match per patient, corresponding to either a full or empty state, was kept for the analysis. When a patient had both full and empty state matches, the empty state match was kept for analysis since there are fewer empty state matches than full ones. When a patient had more than one match of the same state, the match with the greatest number of PPG segments was kept. Ultimately, 90 unique patients were included in the final cohort and 16 of these were empty state matches.

Principal components analysis (PCA) and logistic regression using the principal components (also known as principal components regression or PCR) was performed in R version 4.3.1 using the 'stats' package [13]. PCA is a method in which the original features belonging to each observation are transformed into principal components. The principal components are composed of the sum of different weights (also known as loadings) of the original features, and they are constructed in such a way that the first few components contain most of the variation in the data. Principal components can highlight underlying factors in the data that are difficult to appreciate when variables are assessed individually. Moreover, every component is uncorrelated with every other. In this analysis we retained the principal components that contain greater than 80% of the variation in the data for PCR. The PCR was a logistic regression where the principal components are the independent variables, and the outcomes are the full and empty state labels for each observation (threshold for significance $p < 0.05$).

Random forest was performed using the 'randomForest' package [14], also in R. Since random forest is sensitive to unbalanced data, we undersampled the majority state (i.e., the full state) to balance the labels and specified a large number of trees (ntree = 1000) to ensure that the model has adequate exposure to the full state data.

## The computer vision model

Computer vision models are frequently used to analyze waveform data, and these models have been used with success to analyze PPG data by our group [15,16]. In our experience, models must be adapted to the clinical question and characteristics of the data. In this study a vision transformer (ViT) model [17] customized for 1-dimensional time series data was used for binary classification of the images as arising from either the empty or full state. Raw waveform images were generated for each 20-second PPG segment. The images were then split into training and test sets at the level of the patient so that images from patients in the test set were excluded from the training set. The model was trained for 50 epochs with

PLOS Digital Health

a learning rate of 1e-5 (following hyperparameter optimization) and using a cross-entropy loss function. Model performance was assessed using leave-P-out cross validation, where P = 2.

## Results

### Study population and baseline characteristics

Of the 250 patients enrolled in the study, 90 patients experienced the predefined outcomes, and had sufficient data for inclusion in the final study cohort. The patient flow diagram is shown in Fig 2. Since only patients with DSS had shock or received fluid, all patients in the final cohort had DSS. Among these, 29 patients (32%) had at least one episode of recurrent shock, and all patients received fluid therapy during the 24-hour study period. However, since only events associated with a corresponding PPG signal can be used in the analysis, there are fewer empty and full states than recurrent shock and eligible fluid initiation events, respectively.

Baseline characteristics of the 90 patients included in this analysis are shown in Table 1. Among the patients included in this analysis 39% were children under 16 years of age. There were a greater number of females, and most patients were seen 3–6 days following the onset of illness. In general, patients were healthy, without comorbidities, and had a normal body mass index. Bloodwork at the time of enrollment is notable for hemoconcentration, thrombocytopenia and elevated liver enzymes. All patients had shock on admission. With respect to the clinical outcomes 32% had further shock (i.e., recurrent shock) following admission and all patients received colloid or crystalloid fluid treatment. In total there were 181 fluid initiation events of which 55 (30%) were with colloid fluid. For completeness, patient characteristics for the entire cohort, including patients excluded from this analysis, are shown in S3 Table.

### Principal components regression and physiologic phenotypes

A set of 23 features including variability and waveform characteristics were extracted for each observation (corresponding to a unique patient as discussed above). Features for each observation were the mean of values from all of the PPG

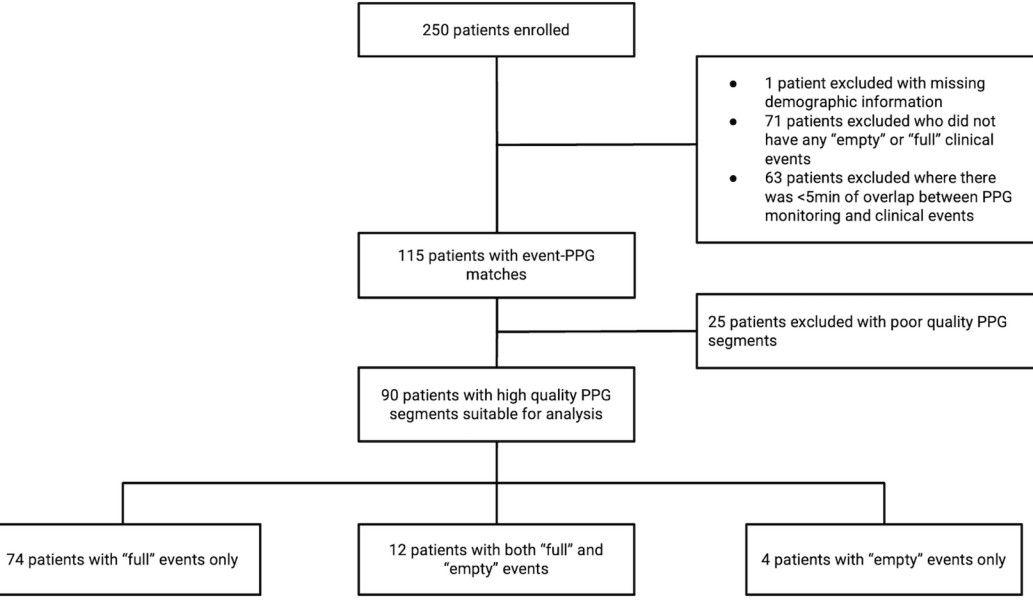

**Fig 2. Flow diagram showing the source of the study population.**

**Table 1. Baseline characteristics for the 90 patients included in the analysis.**

| Characteristics | Median (IQR) or Number (%) (n=90) |
| --- | --- |
| Age, years | 19.0 (12.0–30.8) |
| Children <16 years | 35 (38.9) |
| Adults >=16 years | 55 (61.1) |
| Female gender | 54 (60.0) |
| Body mass index, kg/m2 | 21.7 (17.8–24.4) |
| Day of illness | |
| <3 days | 2 (2.2) |
| 3 to 4 days | 28 (31.1) |
| 5 to 6 days | 60 (66.7) |
| >6 days | None |
| Transferred from another hospital | 40 (44.4) |
| Comorbidities | |
| Hypertension | 1 (1.1) |
| Diabetes | 4 (4.4) |
| Physical exam signs and symptoms | |
| Pleural effusion or rales | 30 (33.3) |
| Bruising or petechiae | 78 (86.7) |
| Headache | 51 (56.7) |
| Vomiting | 47 (52.2) |
| Diarrhea | 23 (25.6) |
| Abdominal pain | 64 (71.1) |
| Bleeding symptoms | 16 (17.8) |
| Respiratory symptoms | 18 (20.0) |
| Vital signs on enrollment | |
| Temperature | 37.0 (37.0–37.0) |
| Heart rate | 95.5 (84.2–107.8) |
| Systolic blood pressure | 104.5 (100.0–110.0) |
| Diastolic blood pressure | 80.0 (70.0–80.0) |
| Pulse pressure | 30.0 (20.0–30.0) |
| Respiratory rate | 22.5 (20.0–26.0) |
| Laboratory values on enrollment | |
| White blood cells (k/uL) | 4.6 (3.1–6.1) |
| Hematocrit (%) | 47.2 (43.6–51.1) |
| Platelets (k/uL) | 22.0 (12.2–33.8) |
| AST (U/L) | 145.0 (89.8–337.2) |
| ALT (U/L) | 76.5 (38.0–195.6) |
| Creatinine (U/L) | 53.5 (49.2–71.0) |

segments for that observation. PCA was applied to the features to determine principal components. In this analysis, principal components 1 through to 4 contain 84% of the variation in the data and were kept for logistic regression.

PCR using the first 4 principal components showed that principal component 1 (PC1, B=0.45, p-value=0.016) and principal component 4 (PC4, B=0.59, p-value=0.036) are positively associated with the empty state. The ROC plot along with the AUC is shown in S3 Fig. The composition of PC1 and PC4 is shown in Fig 3. As shown, for PC1 a variety of features indicating greater heart rate variability, greater waveform variability and broader waveform beats weighed negatively

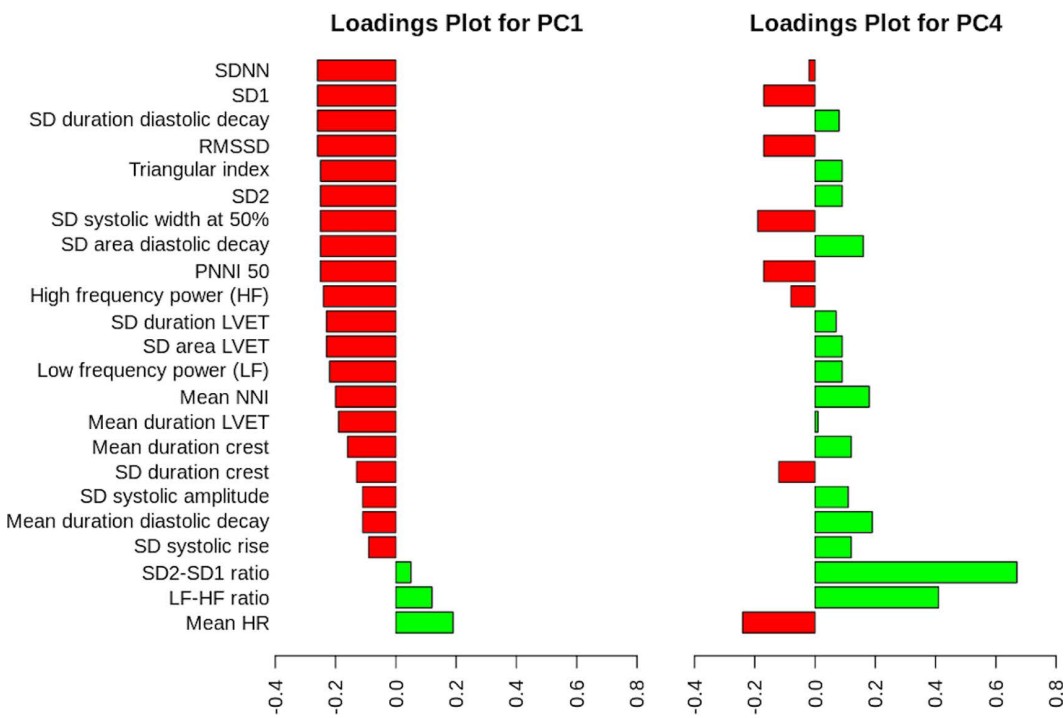

**Fig 3. Loadings plots for PC1 and PC4.** Abbreviations: SD – standard deviation, RMSSD – root mean square successive difference, PNNI 50 – percentage of successive NN intervals that differ by more than 50 ms, NNI – N-N interval, LVET – left ventricular ejection time.

(red bars), while features indicating greater sympathetic activation weighed positively (green bars). For PC4, two features dominated by magnitude, the SD2-SD1 ratio and the LF-HF ratio. The SD2-SD1 ratio is also known as the cardiac sympathetic index. It is the ratio of longer-term variability over short term beat-to-beat variability and it represents sympathetic activation. The LF-HF ratio represents the ratio of low to high frequency power in the waveform. Higher LF-HF values are also associated with sympathetic dominance.

### Random forest and feature importance

We also investigated the contribution of the original features using the random forest method. Since many of the features are interdependent and highly correlated (S2 Table) we remove the most correlated features, with Pearson correlation coefficient > 0.7, for this part of the analysis. The eight features retained were: LF-HF ratio, mean HR, RMSSD, SD duration crest, SD duration LVET, SD2-SD1 ratio, mean duration crest and SD systolic rise. Retained features were chosen based on clinical relevance. The model has an out-of-bag error rate of 33% which corresponds to 67% accuracy in classifying the states of individual patients. The ROC plot along with the AUC is shown in S2 Fig. Feature importance was assessed using the mean decrease in accuracy and in Gini. As shown in Fig 4, the LF-HF ratio and the mean HR were consistently identified as the most important features in classifying the full or empty state.

### Performance of a computer vision model

For each observation (patient-state combination) in the test set, the final prediction was generated by averaging the predicted probabilities across all available segments for a particular patient-state combination to determine if the sample was above the threshold for empty state classification. There were 102 patient-state combinations including 12 patients

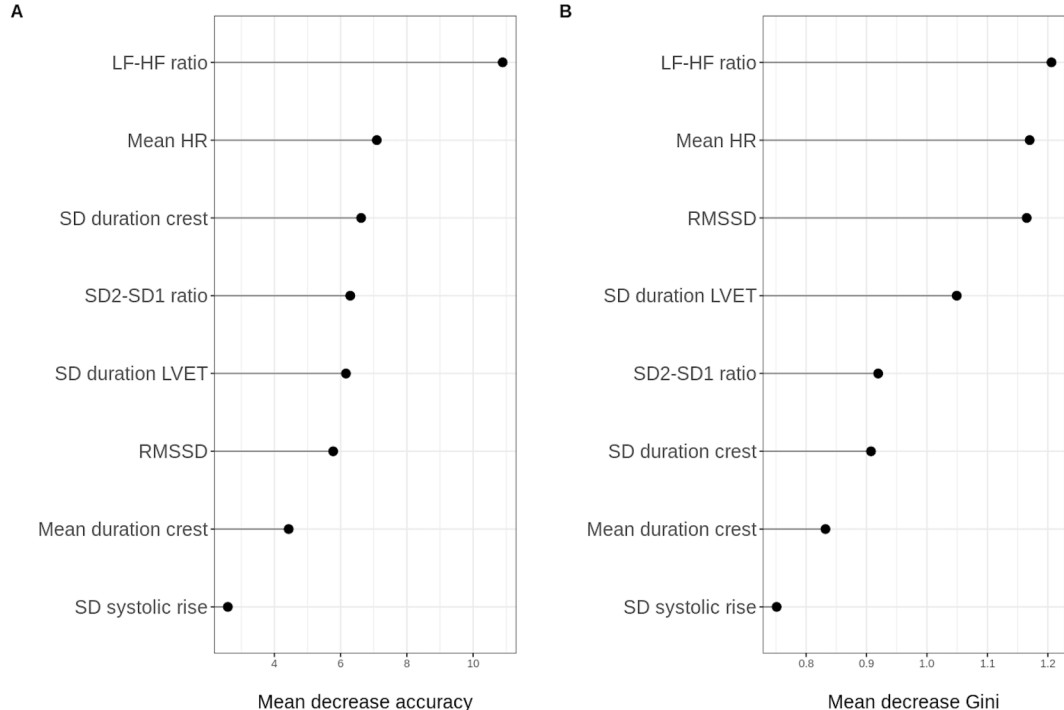

**Fig 4. Feature importance plots.** Abbreviations LF – low frequency power, HF – high frequency power, HR – heart rate, SD – standard deviation, RMSSD – root mean square successive difference.

with both full and empty states. When there were both full and empty states available for a patient, these were treated as separate observations but were always included together in the same test set. Predictions from the vision transformer model were assessed using a threshold which was set to favor prediction of the empty state. We prioritized detection of the empty state because it is associated with further shock. The results of leave-P-out cross validation (P = 2) showed that the model achieved a sensitivity of 0.81 (detection of empty state patients) and a specificity of 0.70 (detection of full state patients) using the selected threshold of separation between the classes. The confusion matrix along with additional metrics are shown in S4 Fig.

## Discussion

In this study we have shown that an artificial intelligence-based approach, including machine learning (PCA/PCR and random forest) and deep learning (computer vision model), provides insight into the underlying physiology of DSS and demonstrates the potential utility of continuous PPG monitoring for volume assessment during DSS. The PCA/PCR shows that PC1 and PC4 are associated with the empty state, suggesting that the empty state is characterized by a physiologic phenotype of sympathetic dominance with diminished variability and narrow waveforms. The random forest, a powerful method for determining individual feature importance, identified the LF-HF ratio and the mean HR as the most important features. Finally, a computer vision model, which was trained on image representations of the raw data, had a sensitivity of 0.81 for the empty state, suggesting that there are waveform biomarkers that can be used to identify low volume states in DSS patients.

Our findings add to the current literature on dengue pathophysiology by highlighting underlying physiologic factors as well as PPG features that differentiate two opposite haemodynamic volume states in DSS. The association of a phenotype of sympathetic dominance with the "empty" state, which was seen both in the principal components and random forest

analyses, is a novel finding. In dengue, increased vascular permeability leads to plasma leakage syndrome [18,19]. At modest levels of intravascular volume loss during the early phase of plasma leakage, increased vascular tone can compensate for volume depletion so that heart rate and blood pressure are unchanged. However, with progressive plasma leakage, compensatory mechanisms can be overwhelmed leading to tachycardia, increased diastolic blood pressure and narrowed pulse pressure [20]. The threshold for decompensation, however, and individual responses to hypovolemia are variable [21]. Furthermore, while increased heart rate is an established predictor for the development of recurrent shock [20] it may manifest late, limiting its usefulness. Therefore, the answers to the questions of when to intervene with fluid and how much fluid should be personalized depending on the risk of decompensation. Our finding that the autonomic response is strongly associated with heamodynamic volume state suggests that tracking autonomic function may be useful adjunct for DSS management.

There are multiple potential benefits of continuous PPG monitoring for the management of severe dengue. First, continuous monitoring would reduce the need for intermittent vital signs assessments, freeing up nursing resources. Second, PPG monitoring is non-invasive. Patients with severe dengue often have profound thrombocytopenia, which increases the risk of bleeding when arterial or central lines are placed for monitoring. Third, PPG monitoring is continuous and the waveform reflects all the important factors – volume, cardiac function and peripheral tone. Indeed, early determination of clinical deterioration is a challenge in DSS because it is a function of multiple factors including intravascular volume, cardiac function, and an individual's ability to compensate [19,22]. Finally, as methods for the analysis of continuous waveform data improve, PPG monitoring will enable a more data driven and dynamic approach to fluid management during DSS that will be easier to standardize across different settings. This has the potential to accelerate research in the field and improve generalizability.

This study has several strengths. To our knowledge it is the first study in which continuous PPG data is aligned with clinical events over a 24-hour period with the goal of classifying different hemodynamic volume states. While Trieu et. al. (2022) recently reported use of the CRI, a PPG-derived measure of clinical deterioration, to predict the occurrence of recurrent shock during DSS, our study uses the PPG waveform to directly interrogate the underlying physiology. Furthermore, we selected analytic methods that are complementary, exploiting the strengths of each approach. We developed a computational pipeline to organize and extract 23 quantitative features from the waveform data, many of which were highly correlated (S2 Table). To address multicollinearity and the need for feature reduction we used PCA/PCR to reduce the number of features to four uncorrelated principal components while preserving most of the information in the data. PCA also enables the identification of underlying factors (manifest via multiple features) that can be difficult to recognize when features are considered independently. Computer vision models which use deep learning are also able to handle non-linear relationships. Moreover, these models can use unstructured PPG waveform data, converted to images, as inputs. While deep learning models are often criticized for being computationally expensive "black boxes", they often demonstrate excellent performance. By interacting with the raw data, a computer vision model potentially captures more information from the data compared to methods that rely on feature engineering. Another strength of our computer vision model is that the performance metrics are based on leave-P-out cross validation (P = 2) with fixed parameters for all folds (30 folds in this case), providing a less biased and more robust estimate of model performance [23].

Our study also has some limitations. First it is a single center study conducted in Vietnam. While adults and children were included, the study cohort was generally homogenous in ethnicity and skin color. Since PPG quality can be impacted by skin tone [24] the results of this study may not be generalizable to populations where there are a different mix of skin colors. We attempted to mitigate this issue by excluding features involving the absolute amplitudes of the beats. The retained features, including the ones that are important in our study, LH-HF and mean HR, depend on the distance between waveform beat peaks or other time-based measures, which are less likely to be impacted by skin tones. However, given that PPG beat amplitudes are associated with intravascular volume [25], excluding these features may reduce model performance. Moreover, since heart rate variability itself is different between races and ethnicities [26], the focus

on variability features in our study may further limit the generalizability of our models. Therefore, additional study among patients with different skin tones and ethnicities is essential to confirm our findings in other populations. Secondly, the "full" and "empty" states are novel concepts that were defined retrospectively and inferred based on nearby clinical events. As a result, the labels are not precise, and each label corresponds to a range of clinical hemodynamic states. For example, patients with multiple episodes of recurrent shock requiring multiple fluid initiation events are clinically different from patients who improve with the first episode of fluid treatment following initial shock, but within our classification scheme both can be assigned the "full" state label following a fluid initiation event. Third, we chose to preprocess the data in the same way (i.e., use the same PPG segments) for all parts of the analysis. Since feature extraction can only be accomplished using very high-quality PPG segments, it was necessary to sample shorter duration PPG segments (20-seconds instead of 1-minute) and to employ a stringent multistep procedure for quality assessment. This reduced the amount of data available for each patient and the number of patients in the final cohort. Since deep learning methods require large amounts of data, this negatively impacted the performance of our computer vision model. Therefore, additional studies using longer PPG segments and more permissive quality procedures should be done to determine the true potential of computer vision models for volume assessment in DSS. Finally, the number of patients in our final cohort (90 patients) is relatively small, especially in relation to the number of PPG features (23 features). We tried to mitigate this in our approach to the analysis. By using PCA for dimension reduction the number of features were reduced from 23 to 4 for the logistic regression. As well, random forest, compared to other methods, demonstrates superior performance on smaller datasets [27]. Nonetheless, the performance of our models needs to be confirmed in larger cohorts.

In summary, this is the first study conducted involving PPG in acute DSS where PPG segments are matched at precise time points to clinical states. By investigating opposite haemodynamic volume states, based on the novel predefined concepts of "empty" and "full", we provided proof of concept that the PPG signals reflect the underlying state in the acute setting. Using an artificial intelligence-based approach, we show that heightened sympathetic drive is an important factor associated with the "empty" state and we develop a computer vision model which shows promising performance using real-world data. We expect that with refinement, machine learning and deep learning models using continuous PPG monitoring could change the way that DSS is managed, enabling dynamic, personalized and precise fluid management.

## Supporting information

**S1 Table.  Description of the HRV and waveform features.**
(DOCX)

**S2 Table.  Correlation table for the HRV and waveform features.**
(DOCX)

**S3 Table.  Baseline characteristics and outcomes.**
(DOCX)

**S1 Fig.  Example patient clinical courses showing how the "empty" and "full" states were defined in relation to clinical events.** Reshock refers to recurrent shock. A An 11-year-old girl admitted to the pediatric ICU, B A 27-year-old woman transferred from an outside hospital, C A 17-year-old woman transferred from an outside hospital.
(DOCX)

**S2 Fig.  Diagram of the fiducial points and waveform features.**
(DOCX)

**S3 Fig.  ROC-AUC plots for the PCR and random forest.**
(DOCX)

**S4 Fig. Confusion matrix and metrics for the vision transformer model.**
(DOCX)

## Acknowledgments

Vietnam ICU Translational Applications Laboratory (VITAL) is a unique multidisciplinary collaboration (vital.oucru.org), funded by the Wellcome Trust as part of its Innovations for Impact strategy.

VITAL Investigators are as follows:

OUCRU inclusive authorship list in Vietnam (alphabetic order by surname): Dang Trung Kien, Dong Huu Khanh Trinh, Joseph Donovan, Du Hong Duc, Ronald Geskus, Ho Bich Hai, Ho Quang Chanh, Ho Van Hien, Hoang Minh Tu Van, Huynh Trung Trieu, Evelyne Kestelyn, Lam Minh Yen, Le Dinh Van Khoa, Le Nguyen Thanh Nhan, Luu Phuoc An, Nguyen Lam Vuong, Nguyen Than Ha Quyen, Nguyen Thi Le Thanh, Nguyen Thi Phuong Dung, Ninh Thi Thanh Van, Phan Nguyen Quoc Khanh, Phung Khanh Lam, Phung Tran Huy Nhat, Guy Thwaites, Louise Thwaites, Tran Minh Duc, Trinh Manh Hung, Hugo Turner, Jennifer Ilo Van Nuil, Sophie Yacoub.

Hospital for Tropical Diseases, Ho Chi Minh City (alphabetic order by surname): Cao Thi Tam, Duong Bich Thuy, Ha Thi Hai Duong, Ho Dang Trung Nghia, Le Buu Chau, Luong Thi Hue Tai, Nguyen Hoan Phu, Nguyen Quoc Viet, Nguyen Thanh Nguyen, Nguyen Thanh Phong, Nguyen Thi Kim Anh, Nguyen Van Hao, Nguyen Van Thanh Duoc, Nguyen Van Vinh Chau, Pham Kieu Nguyet Oanh, Phan Tu Qui, Phan Vinh Tho.

University of Oxford (alphabetic order by surname): David Clifton, Mike English, Heloise Greeff, Huiqi Lu, Jacob McKnight, Chris Paton.Imperial College London (alphabetic order by surname): Pantellis Georgiou, Bernard Hernandez Perez, Kerri Hill-Cawthorne, Alison Holmes, Stefan Karolcik, Damien Ming, Nicolas Moser, Jesus Rodriguez Manzano.

King's College London (alphabetic order by surname): Alberto Gomez, Hamideh Kerdegari, Marc Modat, Reza Razavi.

ETH Zurich (alphabetic order by surname): Abhilash Guru Dutt, Walter Karlen, Michaela Verling, Elias Wicki.

The University of Melbourne (alphabetic order by surname): Linda Denehy, Thomas Rollinson.

## Author contributions

**Conceptualization:** Ngan Nguyen Lyle, Ho Quang Chanh, James Anibal.

**Data curation:** Ngan Nguyen Lyle.

**Formal analysis:** Ngan Nguyen Lyle, Ho Quang Chanh, James Anibal, Stefan Karolcik, Damien Ming, Hai Ho Bich, Khoa Le Dinh Van.

**Funding acquisition:** Pantelis Georgiou, Louise Thwaites, Sophie Yacoub.

**Investigation:** Ho Quang Chanh, Giang Nguyen Thi, Huyen Vu Ngo Thanh, Huy Nguyen Quang, Van Hoang Minh Tu, Khanh Phan Nguyen Quoc, Huynh Trung Trieu.

**Methodology:** Ngan Nguyen Lyle, Ho Quang Chanh, James Anibal, Huy Nguyen Quang, Khoa Le Dinh Van, Louise Thwaites, Sophie Yacoub.

**Project administration:** Hao Nguyen Van, Louise Thwaites, Sophie Yacoub.

**Resources:** Hao Nguyen Van, Qui Tu Phan, Tho Phan Vinh, Tai Luong Thi Hue, Pantelis Georgiou, Louise Thwaites, Sophie Yacoub.

**Supervision:** Pantelis Georgiou, Louise Thwaites, Sophie Yacoub.

**Writing – original draft:** Ngan Nguyen Lyle, Ho Quang Chanh, James Anibal.

**Writing – review & editing:** Ngan Nguyen Lyle, Ho Quang Chanh, Louise Thwaites, Sophie Yacoub.

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
