## [Decision Letter · Decision Letter 0]

PDIG-D-24-00435An artificial intelligence-based approach to identifyvolume status in patients with severe dengue using wearable PPG dataPLOS Digital Health Dear Dr. Lyle, Thank you for submitting your manuscript to PLOS Digital Health. After careful consideration, we feel that it has merit but does not fully meet PLOS Digital Health's publication criteria as it currently stands. Therefore, we invite you to submit a revised version of the manuscript that addresses the points raised during the review process. Please submit your revised manuscript within 60 days Feb 18 2025 11:59PM. If you will need more time than this to complete your revisions, please reply to this message or contact the journal office at digitalhealth@plos.org. Please include the following items when submitting your revised manuscript:* A rebuttal letter that responds to each point raised by the editor and reviewer(s). You should upload this letter as a separate file labeled 'Response to Reviewers '. This file does not need to include responses to any formatting updates and technical items listed in the 'Journal Requirements' section below.* A marked-up copy of your manuscript that highlights changes made to the original version. You should upload this as a separate file labeled 'Revised Manuscript with Track Changes '.* An unmarked version of your revised paper without tracked changes. You should upload this as a separate file labeled 'Manuscript '. If you would like to make changes to your financial disclosure, competing interests statement, or data availability statement, please make these updates within the submission form at the time of resubmission. Guidelines for resubmitting your figure files are available below the reviewer comments at the end of this letter. We look forward to receiving your revised manuscript. Kind regards, Saptarshi Purkayastha, Ph.D.Academic EditorPLOS Digital Health Saptarshi PurkayasthaAcademic EditorPLOS Digital Health Leo Anthony CeliEditor-in-ChiefPLOS Digital Healthorcid.org/0000-0001-6712-6626 **Journal Requirements:**

1. We ask that a manuscript source file is provided at Revision. Please upload your manuscript file as a .doc, .docx, .rtf or .tex.

2. We noticed that you used "unpublished data" in the manuscript. We do not allow these references, as the PLOS data access policy requires that all data be either published with the manuscript or made available in a publicly accessible database. Please amend the supplementary material to include the referenced data or remove the references.

 **Additional Editor Comments (if provided):** Thank you for submitting your manuscript on AI-based monitoring of volume status in severe dengue patients using wearable photoplethysmography (PPG). We have now received comments from three expert reviewers. While they find your work innovative and potentially impactful, they have identified several important areas that require attention before the manuscript can be considered for publication.

Key Areas Requiring Revision:

1. Study Design and Clinical Framing - The binary classification approach needs substantial revision. As Reviewer 2 notes, the current framing of "before vs. after shock" has limited clinical utility. Please reframe the analysis to address more clinically relevant questions, such as predicting future shock events. Include analysis of the excluded patients (n=159) who did not experience shock events, as these contain valuable physiological information for comparison.

2. Statistical Analysis and Methodology - Address the sample size limitations and justify the use of advanced ML methods given the relatively small dataset. Clarify which definitions and choices were pre-specified (e.g., full/empty criteria, signal quality metrics).

3. Claims and Interpretations - Revise statements about the algorithm's clinical utility. Provide evidence-based justification for claims about enhanced clinical care and timely interventions.

Please provide a point-by-point response to all reviewer comments along with your revised manuscript. Take particular care to address the major concerns regarding study design, statistical analysis, and clinical utility.**Reviewers' Comments:** Reviewer's Responses to Questions

**Comments to the Author**

1. Does this manuscript meet PLOS Digital Health’s publication criteria ? Is the manuscript technically sound, and do the data support the conclusions? The manuscript must describe methodologically and ethically rigorous research with conclusions that are appropriately drawn based on the data presented.

Reviewer #1: Yes

Reviewer #2: No

Reviewer #3: Yes

2. Has the statistical analysis been performed appropriately and rigorously?

Reviewer #1: No

Reviewer #2: No

Reviewer #3: Yes

3. Have the authors made all data underlying the findings in their manuscript fully available (please refer to the Data Availability Statement at the start of the manuscript PDF file)?

Reviewer #1: Yes

Reviewer #2: No

Reviewer #3: No

4. Is the manuscript presented in an intelligible fashion and written in standard English?

Reviewer #1: Yes

Reviewer #2: Yes

Reviewer #3: Yes

5. Review Comments to the Author

Reviewer #1: PDIG-D-24-00435

Summary

The manuscript explores an AI-based approach to monitor volume status in severe dengue patients using wearable photoplethysmography (PPG). It presents a prospective study of 250 patients, using machine learning techniques, such as principal components analysis (PCA), random forest, and a computer vision model. The study aims to classify "full" and "empty" hemodynamic states based on PPG data. Promising results include an 81% sensitivity for predicting the empty state using a vision transformer model and identification of sympathetic activation as a key indicator of volume depletion. This work underscores the potential of AI and wearable technology in dynamic and personalized fluid management in dengue shock syndrome (DSS).

Main Contribution

The manuscript makes several important contributions:

1. It provides a novel application of PPG data and AI models in a clinical setting, focusing on volume status monitoring in DSS patients.

2. It highlights key physiologic markers of DSS, including sympathetic activity, and successfully applies both traditional machine learning (PCA and random forest) and deep learning approaches (vision transformers).

3. The study demonstrates the feasibility of using low-cost, non-invasive wearable technology for critical care in resource-constrained settings.

Statistical Analysis Remarks

The following methods were applied:

1. Principal Components Analysis (PCA): PCA was used effectively to reduce dimensionality and highlight key physiologic factors (PC1 and PC4) linked to hemodynamic states.

2. Random Forest: This method identified important features (e.g., LF-HF ratio, mean heart rate) but had an out-of-bag error rate of 33%, limiting predictive reliability.

3. Computer Vision: The vision transformer model showed robust performance (81% sensitivity), though the F1 score (0.47) indicates challenges in handling imbalanced data. It will be beneficial to reference other vision models

Major Corrections Needed

1. Statistical Methods: Although the statistical methods employed are appropriate for the study’s objectives, it will be beneficial to reference other ML /AI models that can be rekevant for the analysis and their role n previous related studies.

2. Definition of Clinical States: The "full" and "empty" volume states are retrospectively defined and appear imprecise. A more granular classification or validation against additional clinical markers (e.g., echocardiography) could be used or at least refrenced.

3. Data Limitations: The study’s generalizability is limited by the homogeneous population (ethnicity and skin tone) and small sample size (90 patients). Addressing this limitation in future work should be emphasized more strongly. Can other methods be applied to strengthen the analysis (e.g.,

4. Feature Selection: While the exclusion of absolute amplitude features is justified, further explanation on how retained features (e.g., LF-HF ratio) avoid such biases would strengthen the methodology. Did the authors used/could use the random forest feature importance/ SHAP/ LIME/ Other XAI methods?

5. Data Quality and Preprocessing: The stringent quality control on PPG data led to reduced sample size. Exploring alternative preprocessing approaches to include more data could enhance model performance.

The manuscript presents innovative research, but addressing these limitations will significantly enhance its impact and reliability.

Reviewer #2: This manuscript describes the use of 24 hour PPG monitoring to assess volume status in Dengue patients, a subset of whom had a “shock” event. The authors report some relationship of extracted features from the PPG device with the clinical labels. The findings are presented as proof of concept, which seems appropriate for the trial design and the modest accuracy.

I have the following comments:

MAJOR:

1) I don’t follow the logic of a binary classifier that aims to separate physiological “state” before vs after a shock event. There is not a clinical circumstance in which I am asking, “is this patient current state either before or after a shock event that was treated with fluids?”. The authors set this problem up as one of monitoring fluid status, which seems appropriate in so far as predicting future events (not asking am I before or after an intervention). Examples of how this can be framed include “PPG features can distinguish individuals who will, vs will not, have a shock event in the next 24 hours”, or some other clinically relevant framing. As such, it is challenging to understand why the reader should be concerned with separability of before vs after shock events. At a minimum, the suggestion is to analyze all pre-shock event physiology to see if such a discriminator is possible - and if not, would severely limit the interpretation.

2) I am concerned that the sample size here is inadequate to support anything other than descriptive statistics (rather than advanced ML)

3) The authors seem to mis-construe the inferences based on their framing and on the “sensitivity”, by stating in the discussion that 81% sensitivity means the method has utility in predicting future shock events. As I understand the framing, the algorithm does predict future shock events. Even if it did (per suggestion in comment #1 for new analysis, the text about sensitivity seems to be conflating PPV with sensitivity. By their own dichotomous report, where specificity is lower value (even though the post-shock full stats is the not the “negative” of a pre-shock empty state), a test with higher sensitivity than specificity would tend to have better rule-out, than rule-in, performance (although the prior is the main determinant obviously).

4) The authors speculate that PPG might somehow be “more informative” than clinical assessment of volume - this seems like quite a stretch - no attempt was made to compare the PPG results with clinical predictions here. At best, the authors might speculate that some PPG features could be used as adjunct to clinical assessment (rather than jump all the way to PPG being more informative - no comparison was made of, say, PPG vs clinical assessment).

5) I don’t understand the claim that PPG somehow frees up nursing resources due to taking vitals less frequently - again, seems like a large stretch in many respects - if nurses are even taking vital signs (vs some other support staff plus telemetry if the person is sick enough to worry about shock), this feels like a minority of a shift’s burden, and given the core necessity of vitals in critical care, it seems implausible vitals will ever be fore-gone in favor of some black box algorithm?

6) The discussion speculation is unjustified: there is no evidence in the report that the algorithm is accurate enough to enhance clinical care, or to make fluid interventions more “timely”. Again, the prediction was not set up in this paper, as far as I understand, as one of “will this patient have a shock event or not”. Instead it was set up as “is this patient before or after a shock event followed by a successful fluid intervention”. Even if the authors do this, they would presumably need to march through clinical examples taking FP and FN rates and risks into account to justify questions like "how accurate would this need to be for clinical implementation?".

7) Most patients in this cohort (n=159 it seems) were excluded, a large portion of which seem did not have a shock event - but these subjects presumably carry important physiological information and should be analyzed. It seems for example very important to compare the PPG features of those excluded (no shock) individuals against those who did have a shock event (the “empty” cohort). This would greatly strengthen the story, and align with the proposed importance (a future clinical prediction tool to aid in clinical care).

8) Related to the above, some additional clarifications will help the labeling. For example, “Full” state is not just fluid repleted, but also clinically improved. So, it seems, we have patients without shock events (presumably none got fluids?), and then we have those with shock events (presumably all got fluids then?), and of those, some that recur or are unstable / poorly responsive. Seems it would be very helpful to chart out the distribution of these possibilities, which will also se up more obviously when and how PPG based fluid status might be useful in clinical flow.

9) Please indicate which definitions and choices were pre-specified (full and empty criteria? Signal quality metrics? etc).

MINOR:

1) I assume the LF-HF ratio was computed from segments longer than 20 seconds. This should be explained. For example in the supplemental table, indicate for each metric the time window used to compute the feature.

2) Acronyms should be explained in all cases (e.g. loading plots, feature abbreviations), in the main text (not just in the supplement).

3) The “discarded” rules are not well articulated - at a minimum, the authors should indicate the percent of PPG signals not meeting the “discard” criteria.

4) Suggest indicating in the correlation matrix whether the individual points are being correlated or whether a mean was taken per subject and the correlation is done at the subject level (the latter of which will often have much higher R values).

5) I do not understand the authors claiming sympathetic activity in shock is a novel finding - seems like a classic mechanism?

6) This bold claim would seem to require some citation(s): “PPG monitoring provides a more dynamic and complete picture of the haemodynamic volume state compared to other methods  like echocardiography.” It is insufficient to simply remark that your team has anecdotal observations on this.

7) I am unclear on the timing of the PPG vs the shock events. Was the empty state really “any” time before shock? Suggest making a figure or some description of these temporal distributions. For example, you could have 90 rows, where X axis is time in days from enrollment, with the time of the PPG indicated, and the time of shock indicated, maybe color shading of when you say the empty vs full periods are.

8) I don’t understand Suppl. Fig 1. It appears the authors have simply ordered the p(state) for empty or full - what is the meaning of such a plot?

Reviewer #3: Administrative Comments:

1. This manuscript meets PLOS Digital Health’s publication criteria.

2. Appropriate statistical analysis is performed.

3. The data is not made available publically due to patient privacy concerns but can be made available on contacting authors.

4. The manuscript is presented well and in standard English.

Review Comments:

Summary:

This study emphasizes the critical need for more reliable and cost-effective methods of fluid management in patients with Dengue Shock Syndrome (DSS), where precise fluid regulation is essential for positive outcomes. The research leverages photoplethysmography (PPG) device monitoring to analyze waveform morphology and develop a DSS-specific algorithm.

The study's methodology is robust and thoughtfully designed, demonstrating a strong potential to address existing challenges in fluid management for DSS. The development of a DSS-specific algorithm is a key strength, as it could streamline clinical workflows and reduce the burden on paramedical staff. Additionally, the study highlights its limitations, such as the need for larger-scale validation or potential variability in PPG signal quality, which strengthens the credibility of its findings. The data is not made available publically due to patient privacy concerns but can be made available on contact.

Overall, this is a well-conducted study that makes a meaningful contribution to the field. Its findings pave the way for future research, particularly in refining the algorithm and exploring its implementation in diverse clinical settings.

Few of these comments below can help further enhance the manuscript.

Major comments:

1. The concepts of “empty state” and “full state” are central to this study. Initially, these terms give the impression of being interdependent and occurring in pairs during an "event." However, it becomes clear later in the manuscript that they are independent variables and can be treated separately. To reduce potential confusion, it would be helpful to clarify early in the manuscript that these states do not necessarily occur in pairs for a shock event.

Additionally, these concepts appear to be novel and are uniquely defined or inferred based on adjacent clinical events. Highlighting their novelty upfront would provide readers with a clearer understanding of their importance and contribution to the study's framework.

2. The decision to exclude absolute amplitude values, such as the height of the systolic peak or the area under the beat, due to the homogeneity of the study population (single skin color) is a thoughtful attempt to address potential biases. However, it would be valuable to discuss whether this exclusion might introduce any limitations.

For instance, while using variation measures (e.g., the standard deviation of systolic peak heights) offers consistency, will it reduce sensitivity to any critical variables? Per research, we know that a decrease in pulse amplitude often serves as an early indicator of impaired circulation or shock. Could this focus on variation measures lead to the algorithm overlooking some of these early, critical signs? Additionally, will this approach risk overfitting to the specific characteristics of the study population or experimental conditions? Discussing these potential trade-offs would strengthen the manuscript and provide a more comprehensive understanding of the methodology’s implications.

3. Given the limited number of studies exploring the use of CRI to predict recurrent shock in DSS patients, it would be valuable to include an explanation of how this new algorithm sets itself apart from the features used to calculate CRI.

Minor comments

1. The low sample size for the "empty" population in the training set seems like it might be a limitation. It would be benificial to expand on how this might impact the generalizability of the findings? It would be great to understand if there are ways this was addressed or if it opens up opportunities for future work.

2. The term "explainable" dengue-specific algorithm is mentioned in the introduction, and it would be helpful to clarify what "explainable" means in this context. Does it suggest that the algorithm will provide doctors with insights into why a particular status is determined for a subject? If so, it would be interesting to learn more about how this "explanation" is incorporated into the algorithm’s design.

3. It would be nice to include details about the envisioned practical application for this algorithm. For instance, could it be integrated into bedside tablets or other monitoring systems? A brief description of how it might be used in clinical settings would provide helpful context for its potential impact.

Overall, this study presents a promising approach to fluid management in DSS patients, and the development of a DSS-specific algorithm is an exciting contribution to the field. Addressing the points mentioned above would further strengthen the manuscript and provide additional clarity for readers.

6. PLOS authors have the option to publish the peer review history of their article (what does this mean? ). If published, this will include your full peer review and any attached files.

**Do you want your identity to be public for this peer review?** For information about this choice, including consent withdrawal, please see our Privacy Policy .

Reviewer #1: No

Reviewer #2: No

Reviewer #3: **Yes: ** Parvati Naliyatthaliyazchayil

---

## [Decision Letter · Decision Letter 1]

PDIG-D-24-00435R1An artificial intelligence-based approach to identify volume status in patients with severe dengue using wearable PPG dataPLOS Digital Health Dear Dr. Lyle, Thank you for submitting your manuscript to PLOS Digital Health. After careful consideration, we feel that it has merit but does not fully meet PLOS Digital Health's publication criteria as it currently stands. Therefore, we invite you to submit a revised version of the manuscript that addresses the points raised during the review process. Please submit your revised manuscript within 30 days Apr 24 2025 11:59PM. If you will need more time than this to complete your revisions, please reply to this message or contact the journal office at digitalhealth@plos.org. Please include the following items when submitting your revised manuscript:* A rebuttal letter that responds to each point raised by the editor and reviewer(s). You should upload this letter as a separate file labeled 'Response to Reviewers '. This file does not need to include responses to any formatting updates and technical items listed in the 'Journal Requirements' section below.* A marked-up copy of your manuscript that highlights changes made to the original version. You should upload this as a separate file labeled 'Revised Manuscript with Track Changes '.* An unmarked version of your revised paper without tracked changes. You should upload this as a separate file labeled 'Manuscript '. If you would like to make changes to your financial disclosure, competing interests statement, or data availability statement, please make these updates within the submission form at the time of resubmission. Guidelines for resubmitting your figure files are available below the reviewer comments at the end of this letter. We look forward to receiving your revised manuscript. Kind regards, Saptarshi Purkayastha, Ph.D.Academic EditorPLOS Digital Health Saptarshi PurkayasthaAcademic EditorPLOS Digital Health Leo Anthony CeliEditor-in-ChiefPLOS Digital Healthorcid.org/0000-0001-6712-6626  **Journal Requirements:** **Additional Editor Comments (if provided):** The changes look appropriate. Please address the reviewer comments regarding binary classification and clarifying a few additional details.**Reviewers' Comments:** Reviewer's Responses to Questions

**Comments to the Author**

1. If the authors have adequately addressed your comments raised in a previous round of review and you feel that this manuscript is now acceptable for publication, you may indicate that here to bypass the “Comments to the Author” section, enter your conflict of interest statement in the “Confidential to Editor” section, and submit your "Accept" recommendation.

Reviewer #2: (No Response)

Reviewer #3: All comments have been addressed

2. Does this manuscript meet PLOS Digital Health’s publication criteria ? Is the manuscript technically sound, and do the data support the conclusions? The manuscript must describe methodologically and ethically rigorous research with conclusions that are appropriately drawn based on the data presented.

Reviewer #2: Yes

Reviewer #3: Yes

3. Has the statistical analysis been performed appropriately and rigorously?

Reviewer #2: Yes

Reviewer #3: Yes

4. Have the authors made all data underlying the findings in their manuscript fully available (please refer to the Data Availability Statement at the start of the manuscript PDF file)?

Reviewer #2: (No Response)

Reviewer #3: No

5. Is the manuscript presented in an intelligible fashion and written in standard English?

Reviewer #2: Yes

Reviewer #3: Yes

6. Review Comments to the Author

Reviewer #2: Thank you for the opportunity to review the revised manuscript, which is improved. I have the following remaining comments.

1) regarding the binary classification, I am still a bit confused. The authors state that “empty” is always before resuscitation, and “full” is always after a shock event (I think we all agree here). Yet they also say in their response, quote: “our study does not classify states “before” vs “after” a shock event.” These seem incompatible statements. The language feels relevant to my comment #2.

2) It feels like the most relevant comparison for your goals seems to be shock vs no shock. If the authors are unwilling to analyze the features from the no-shock group (n=71, it seems, see next comment), it feels like it should be called out directly in the discussion/limitations, that this classifier framing for proof-of-concept is distinct in nature from what is desired in the future: that it will need to run with inputs from pre-shock data, to bridge the gap from this work to a future clinical implementation (prediction) task. As written, especially with the emphasis on sensitivity of 81 (without even calling out the specificity - which by the way is listed as 67 in abstract, but 70 in the paper, and needs to be reconciled please), makes it seem as though this comparison is actually the desired one for future implementation.

3) My comment #7 did not say most subjects did not have shock. It said most were excluded (159, per your diagram, is more than have of consented total), and of those, “a large portion” did not have shock (n=71). This is pertaining to comment #2, which refers to the question, why not analyze those 71 for the same features, to see if those who did go into shock are different than those who didn’t? Seems highly relevant, and would enormously improve the impact of this work.

4) regarding the sample size, the authors have quoted a paper (Do We Need Hundreds of Classifiers to Solve Real World Classification Problems?) to indicate small sample sizes are acceptable. However, this is not the conclusion of that paper, to my read, which defined “maximum accuracy” as the top accuracy across many classifiers applied to a single data set. This is a very different question from what is at stake here: is the sample size sufficient to make inferences about the problem at hand. This is not simply a statistical question applied indiscriminately (although it is often framed as such), it depends on the complexity of the input signals or features, the uncertainty in the labels, among many other factors. My suggestion is to call out the small sample size in the limitations.

5) the authors responded to my question that the empty state was not “any” 2 hours period - but their methods continue to read, and I quote: “An “empty” state was any two-hour period before a shock event.” (Page 6, line 137, in the marked version). Please amend the methods to align with your response to my question.

Reviewer #3: Administrative-

1. If the authors have adequately addressed your comments raised in a previous round of review - Yes, all comments have been addressed and necessary updates made.

2. Does this manuscript meet PLOS Digital Health’s publication criteria? - Yes

3. Has the statistical analysis been performed appropriately and rigorously?- Yes

4. Have the authors made all data underlying the findings in their manuscript fully available - Patient data is not made available due to Patient confidentiality and is available on request

Final comments-

Thank you for the updated manuscript. The Authors have made the necessary updates to their manuscript based on the my review comments. I believe that this study greatly adds to the exsiting body of research in DSS fuild managment doamain.

7. PLOS authors have the option to publish the peer review history of their article (what does this mean? ). If published, this will include your full peer review and any attached files.

**Do you want your identity to be public for this peer review?** For information about this choice, including consent withdrawal, please see our Privacy Policy .

Reviewer #2: No

Reviewer #3: **Yes: ** Parvati Naliyatthaliyazchayil

---

## [Decision Letter · Decision Letter 2]

An artificial intelligence-based approach to identify volume status in patients with severe dengue using wearable PPG data

PDIG-D-24-00435R2

Dear Dr. Lyle,

We are pleased to inform you that your manuscript 'An artificial intelligence-based approach to identify volume status in patients with severe dengue using wearable PPG data' has been provisionally accepted for publication in PLOS Digital Health.

Best regards,

Daniel B. Forger

Section Editor

PLOS Digital Health

**Additional Editor Comments (if provided):**

**Reviewer Comments (if any, and for reference):**

Reviewer's Responses to Questions

**Comments to the Author**

1. If the authors have adequately addressed your comments raised in a previous round of review and you feel that this manuscript is now acceptable for publication, you may indicate that here to bypass the “Comments to the Author” section, enter your conflict of interest statement in the “Confidential to Editor” section, and submit your "Accept" recommendation.

Reviewer #2: All comments have been addressed

2. Does this manuscript meet PLOS Digital Health’s publication criteria ? Is the manuscript technically sound, and do the data support the conclusions? The manuscript must describe methodologically and ethically rigorous research with conclusions that are appropriately drawn based on the data presented.

Reviewer #2: Yes

3. Has the statistical analysis been performed appropriately and rigorously?

Reviewer #2: Yes

4. Have the authors made all data underlying the findings in their manuscript fully available (please refer to the Data Availability Statement at the start of the manuscript PDF file)?

Reviewer #2: Yes

5. Is the manuscript presented in an intelligible fashion and written in standard English?

Reviewer #2: Yes

6. Review Comments to the Author

Reviewer #2: the authors have addressed my comments.

7. PLOS authors have the option to publish the peer review history of their article (what does this mean? ). If published, this will include your full peer review and any attached files.

**Do you want your identity to be public for this peer review?** For information about this choice, including consent withdrawal, please see our Privacy Policy .

Reviewer #2: No
